# Joint Representation Learning for Retrieval and Annotation of Genomic Interval Sets

**DOI:** 10.3390/bioengineering11030263

**Published:** 2024-03-08

**Authors:** Erfaneh Gharavi, Nathan J. LeRoy, Guangtao Zheng, Aidong Zhang, Donald E. Brown, Nathan C. Sheffield

**Affiliations:** 1Center for Public Health Genomics, School of Medicine, University of Virginia, Charlottesville, VA 22908, USA; 2School of Data Science, University of Virginia, Charlottesville, VA 22904, USA; 3Department of Biomedical Engineering, School of Medicine, University of Virginia, Charlottesville, VA 22904, USA; 4Department of Computer Science, School of Engineering, University of Virginia, Charlottesville, VA 22908, USA; 5Department of Systems and Information Engineering, University of Virginia, Charlottesville, VA 22908, USA; 6Department of Public Health Sciences, School of Medicine, University of Virginia, Charlottesville, VA 22908, USA; 7Department of Biochemistry and Molecular Genetics, School of Medicine, University of Virginia, Charlottesville, VA 22908, USA; 8Child Health Research Center, School of Medicine, University of Virginia, Charlottesville, VA 22908, USA

**Keywords:** genomic intervals, search, metadata, embeddings, representation learning, information retrieval, functional genomics, computational genomics, chromatin

## Abstract

As available genomic interval data increase in scale, we require fast systems to search them. A common approach is simple string matching to compare a search term to metadata, but this is limited by incomplete or inaccurate annotations. An alternative is to compare data directly through genomic region overlap analysis, but this approach leads to challenges like sparsity, high dimensionality, and computational expense. We require novel methods to quickly and flexibly query large, messy genomic interval databases. Here, we develop a genomic interval search system using representation learning. We train numerical embeddings for a collection of region sets simultaneously with their metadata labels, capturing similarity between region sets and their metadata in a low-dimensional space. Using these learned co-embeddings, we develop a system that solves three related information retrieval tasks using embedding distance computations: retrieving region sets related to a user query string, suggesting new labels for database region sets, and retrieving database region sets similar to a query region set. We evaluate these use cases and show that jointly learned representations of region sets and metadata are a promising approach for fast, flexible, and accurate genomic region information retrieval.

## 1. Introduction

The increasing volume of biological data has motivated initiatives to facilitate data search, retrieval, and interoperability [1,2,3,4,5,6,7]. This is particularly important for epigenome data, for which data search and retrieval has become intractable [8]. Two natural ways to search for genomic interval data are: First, to search with natural language for data relevant to a query; or second, to search with a query set of genomic intervals to find similar data. For a natural language search, a typical query system uses string pattern matching to search the metadata annotations in the database [9,10,11]. While such lexical search systems work, they are limited because they rely on annotations rather than the data itself and, therefore, are only as effective as the quality of the annotations. Furthermore, they may miss biological similarities between experiments with different annotations: for example, prostate cancer and breast cancer data have a clear biological relationship [12], yet this may be missed by metadata matching. Finally, metadata matching cannot be used at all for unannotated data.

For the second type of epigenome query, instead of querying using natural language terms, the query consists of genomic intervals. Many methods exist to facilitate this type of analysis [13,14,15,16,17,18,19,20,21]. These methods use the data itself and are powerful and widely used, but they also have limitations: they may not scale to big data, they require complex input, and they fail to take into account the biological context in which the data were generated.

To address limitations of these common analytical approaches, we present an alternative approach that bridges the gap between searching for genomic data based on natural language and searching with interval overlaps. Our approach leverages recent advances in neural embedding methods and the growing corpus of epigenome data to tie natural language to genomic intervals. Neural embedding approaches show great promise for a variety of biological applications [22,23,24,25,26,27,28]. In particular, the StarSpace neural embedding approach has been recently used to learn representations of cancer mutational signatures and has shown to be resource-efficient, flexible, and scalable [29]. Here, we use StarSpace [30] to jointly embed genomic interval region sets with associated metadata into a shared latent embedding space. Using these learned co-embeddings, we develop a system that solves three related information retrieval tasks: first, retrieving region sets related to a user query string; second, suggesting new labels for unannotated database region sets; and third, retrieving database region sets similar to a query region set. Our models do these tasks using lower-dimensional embedding distance computations instead of computing interval overlaps or string matching. We evaluate these use cases and show that jointly learned representations of region sets and metadata are a promising approach for fast, flexible, and accurate genomic region information retrieval. Furthermore, we show how our approach can search data with sparse or inaccurate metadata and can also standardize metadata annotation and correct annotation errors.

## 2. Results

### 2.1. Overview

To learn low-dimensional representations of genomic regions and their associated metadata, we first assembled a dataset of 2548 ChIP-seq region sets from ENCODE [31]. We focused on two metadata attributes—cell type and ChIP antibody target—and limited the dataset to eight activating histone modifications: *H3K4me1*, *H3K4me2*, *H3K4me3*, *H3K27ac*, *H3K9ac*, *H3K9me1*, *H3K79me2*, and *H4K29me1*, which represent 131 cell types (Figure 1A).

Like many natural language methods, StarSpace requires documents to share a vocabulary, which corresponds to a predefined genomic interval “universe” for region set data. After hyper-parameter tuning, we selected a 1000 bp tiling universe and tokenized the raw intervals by mapping each interval to one in the universe (Figure 1B). After the tokenization step, we considered each tokenized region set as a document. We split the data into a test set (15%) and a training set (85%), which we further subdivided into training (90%) and validation sets (10%) for hyper-parameter tuning and early stopping in the training phase (Figure 1C). We then trained a StarSpace model to jointly learn numerical representations for region sets with the cell type and ChIP antibody as labels (Figure 1D; see Methods for details). The StarSpace algorithm converts each region set and its corresponding label to a numerical vector—or embedding, an n-dimensional vector represented in embedding space—putting biologically related region set vectors and their labels close to one another in the shared latent space (Figure 1E).

Next, we evaluated whether the model could capture biological relationships. As a global sanity check, we visualized label embeddings using Uniform Manifold Approximation and Projection (UMAP) in two-dimensional space. We observed results generally in line with expectations, with similar antibodies mapped near one another, such as all the *H3K4* methylation marks (Figure 1F). Cell type labels also showed some grouping by tissue type (Figure 1G), indicating that the embeddings may capture biological information during the training process.

To test the embeddings more rigorously, we next designed experiments in three different scenarios (Figure 1H): First, we provide a query term from a set of antibody or cell type labels as input, and the model outputs a ranked set of region sets similar to the query term. In the second scenario, we provide a region set as input, and the model outputs a set of ranked labels annotating the region set. This scenario tests the capability of working with unannotated data. In the third scenario, we provide a query region set, and the model retrieves similar region sets. We trained three separate StarSpace models: one using only the cell type label, one using only the antibody label, and one using both. In each scenario, we evaluated each of the three models using the test set.

### 2.2. Scenario 1. Retrieve Region Sets for a Metadata Query

Scenario 1 evaluates whether we can accurately retrieve region sets that are related to a query search term. The search term could come from any of the metadata of the region sets in our training set, such as *K562* from cell type labels or *H3K4me2* from antibody labels. To retrieve related region sets, we first map the search term to its numerical vector. Next, we identify the nearest neighbor embeddings of region sets in our model by calculating the cosine similarity between the search term and region set embeddings. We return a ranked list of the most similar region sets (Figure 2A).

To evaluate the performance for Scenario 1, we first assessed the similarity of a predicted region set label to the actual label for each region in the test set. Since the StarSpace algorithm aims to maximize the similarity between the embeddings of the region sets and the embeddings of their labels, a well-trained model should have a high similarity value between each region set and its corresponding label. Overall, the similarities are high, and similarity values for models trained with only one label type (only cell type or only antibody) are higher than those trained with both (Figure 2B). To test the performance of the search system, we used r-precision, which is defined as the fraction of results returned that are relevant, where relevant means its true label matches the provided search term (see Methods). We calculated r-precision separately for each cell type and antibody on each of the three StarSpace models (Figure 2C). The mean r-precision across terms shows that the antibody model performs better than the cell type model (Figure 2D); this result reflects the increased difficulty of the cell type task, since the antibody task has only 8 classes whereas the cell type task has 113. Furthermore, training the StarSpace model on one label separately resulted in better performance, which reflects a similar complexity issue: training on one label reduces the complexity of the model by limiting it to a specific type of metadata. Training separately requires training multiple models and results in multiple embedding vectors for each region set.

To demonstrate a specific example, we visualized the results for query string *22rv1*, the name of a human prostate carcinoma cell line (Figure 2E). The top hit was the only response in the test set with label *22rv1*, indicating that the retrieval task was successful. Other highly scoring region sets include other prostate cancer cell lines, *c4-2b* and *vcap*, as well as *mcf-7*, a breast cancer cell line. Thus, the model reflects known relationships between prostate and breast cancer in the embedding space [12]. For a query term of *H3K4me2*, the model flawlessly ranks the top four region sets with the same label from our test set at the top (Figure 2F). The remaining region sets belong to mono- and tri-methylation of the histone H3 lysine K4, which are similar marks.

To explore global performance trends, we calculated the cosine similarity between each region set embedding in the test dataset with the embeddings of the corresponding labels for either antibody (Figure 2G) or cell type (Appendix A). The model generally retrieves the expected inputs. Furthermore, we observed global relationships among these labels; *H3K4me2* and *H3K4me3* form one cluster, which then joins with *H3K4me1*, *H3K27ac*, and *H3K9ac*; these marks are more distant from *H4K20me1*, *H3K9me1*, and *H3K79me2*. Together, these results suggest that a co-embedding model is a promising approach for a natural language search for relevant biological region sets.

### 2.3. Scenario 2. Annotate Unlabeled Region Sets

Scenario 2 provides annotation suggestions for interval sets with missing or inaccurate metadata information. In this use case, region sets from the test set are provided as queries, and the model suggests annotations. We first tokenize the query region set and then convert the tokenized query set to its embedding representation using the trained StarSpace model. We then compute the cosine similarity between the query region set embedding and label embeddings, returning a ranked list. The label achieving the maximum similarity values (minimum distance) for each label type is proposed as the label for the query region set (Figure 3A).

To demonstrate, we queried with a test region set labeled with *gm12866* and *H3K4me3*. The most similar labels of each type clearly correspond to the correct labels, with top hits including various lymphoblastoid cell lines and the *h3k4me3* antibody label (Figure 3B).

To evaluate performance globally, we used two performance metrics. First, the reciprocal rank (RR) calculates the rank at which the label relevant to the region set is retrieved (see Methods). We consider a label as relevant if it is the same as the actual label of the test region set. RR is 1 if a relevant label is retrieved at rank 1, and RR is 0.5 when a relevant label is retrieved at rank 2. The RR statistic shows that the models have generally good performance; in particular, the antibody-trained models are able to achieve nearly perfect results (Figure 3C). To summarize these results, we computed the mean of the reciprocal ranking for all the query region sets in our test dataset (MRR). For all models and tests, our MRR score is above 0.6, indicating that the relevant result is most frequently ranked as the top or second search result (Figure 3D). The MRR of the model trained on just one label is again superior to the model trained on both labels.

For our second evaluation approach, we consider Scenario 2 as a classification task and use a confusion matrix to explore the results. The confusion matrix reflects generally strong performance and shows that the confusion mainly occurs between *H3K4me3* and *H3K4me2* (Figure 3E). While chemically distinct, these two histone modifications are highly similar, further indicating that the model is capturing information about the underlying biology. In addition, there is some confusion between *H3K4me3* and *H3K9ac*: both known to mark promoters. We also computed the micro-averaged F1 score to measure classification performance (Figure 3F; see Section 4). We achieved good F1 scores across all tasks and saw that the models trained on independent labels again outperformed the model trained on both labels simultaneously. Altogether, these results illustrate a highly capable annotation model that could be a useful automated system for annotating unlabeled or incorrectly annotated genomic interval data with standardized terms.

### 2.4. Scenario 3. Retrieve Region Sets for a Query Region Set

Scenario 3 finds database region sets similar to a given query region set. As before, we first tokenize the query and then convert it to an embedding using the trained StarSpace model. We then calculate the similarity of the vector to the database region set embeddings to retrieve and rank results (Figure 4A). The region set database is our training set, and our queries come from the test set. To evaluate, we define hits as the number of region sets with matching antibody and cell type labels.

To demonstrate, we queried with a region set annotated with *H3K27ac* and *vcap* labels. Using the model that is trained only on the cell types, the two most similar region sets also have *vcap* as the cell type label (Figure 4B). The remaining most similar region sets come from other prostate cancer cell lines: *lncap*, *c4-2b*, and *22rv1*. The antibody results similarly retrieved the same target antibody as the query region set (Figure 4C).

To summarize these results globally, we calculated the similarity of each test region set to all the database (training) region sets. We observed consistently good performance, with higher similarity when the query and dataset region sets share identical or similar labels for both antibody (Figure 4D) and cell type labels (Appendix A). We also compared our results with the Jaccard similarity. We selected files labeled as *k652*, *mcf-7*, or *a549* and calculated the Jaccard similarity of each selected region set to all database region sets with one of these labels. The results shows that the Jaccard similarity is correlated with our embedding similarity score (Appendix A).

### 2.5. Annotating External Data with a Pre-Trained Model

Next, to apply our model to a real large-scale use case, we turned to the task of external data annotation. We used geofetch [32] to collect 32,174 *hg19* BED files with associated metadata from the Gene Expression Omnibus (GEO). Since these datasets come from a variety of sources, the metadata are not standardized. Our goal is to apply the annotation task from Scenario 2, using the pre-trained StarSpace model to propose standardized labels for these new region sets.

Following Scenario 2, we tokenized the GEO regions sets, then used the StarSpace model trained on the ENCODE dataset to project embeddings, and finally ranked the labels from our trained model by similarity to the GEO region set embeddings (Figure 5A). We then annotated each file with the top-ranked label (Table S1). Since this dataset lacks standardized correct answers, to assess performance loosely, we identified GEO files that had labels corresponding to our trained labels. For antibody labels, 21% of the data had a corresponding label, and our model accurately labeled 54% of these samples (Figure 5B). For cell types, 14% of the GEO samples could be mapped to our labels, and of these, 75% were labeled correctly (Figure 5B). For example, sample GSM5614147 (“H3K4me1 1TF-4”) was correctly labeled (Figure 5C); similarly, sample GSM4797835 (“K562 H3K4me2 dmf60 fragments”) was confidently labeled as *K562* (Figure 5D). Looking closer at the incorrect predictions, we noticed that many of these were near misses; for example, the predicted antibody for GSM5455054 (“Primary AFH1 H3K27ac peaks”) was *H3K9ac*, but *H3K27ac* was ranked second with a negligible difference in the similarity score (Figure 5E). An incorrect result for file GSM3223711 (“VCaP shCt AR peaks”) shows that the top four labels were prostate cancer lines, though the correct answer was ranked fourth (Figure 5F). To overcome this issue, we defined a confidence interval between the first and second predicted labels and reported the results as confident if this value passed a 0.1 threshold.

We also often identified spelling mistakes or non-standard representations in filenames or other metadata. We reasoned that this approach could be used to identify such inconsistencies. For example, we found data labeled with *h3k4m3* or *h3k4tri*, which we assume are slight variations of the *H3K4me3* label. To assess this more systematically, we used regular expressions to extract likely incomplete or misspelled labels. We identified several examples for which the model could correct these issues. For example, the model predicted *h3k4me3* as the most similar label for sample GSM2864752 (“IgE H3K3me3 peaks”). Since the provided *H3K3me3* label is likely an error, our model predicts the label for this file should be *h3k4me3* (Figure 5G). Similarly, for sample GSM5220509 (“Lane1 2D H3K27ace BMI1 macs2 peaks”), the model predicts an acetylation mark as the first predicted label (Figure 5H; Appendix A). A variety of similar examples show the utility of this process: our model can update labels *k9ac* to *h3k9ac*, *k27ac* to *h3k27ac*, *K4me3* to *h3k4me3*, *K4ME1* and *K4me1* to *h3k4me1*, *h3k4m3* to *h3k4me3*, *K27* to *h3k27ac*, and *K4* to *h3k4me3* (Appendix A).

Our model predicted *h3k4me3* as the label for two files with “tri” in the labels, and it corrected “mo” and “mono” labels (Appendix A). The model can only predict from the eight labels we used when training, so it is rather limited; however, for region sets with other labels, the retrieved labels typically share biological similarity. For instance, the predicted label for *H3k18ac*—which does not exist in our training set—is *h3k27ac*, which is a biologically similar marker. We found similar results for labels *H4k12me1* and *H3k79me3* (Appendix A). Furthermore, for labels *h3k36me3*, *h3k27me3*, *h3k9me3*, and *h4k20me3*, the similarity to the closest labels is smaller; these are repressive marks, and our model is only trained on eight activating marks, so the model is able to convey that none of the trained labels are a good fit (Appendix A). Taken together, these results confirm that our model can be used for metadata standardization by predicting the same label for similar files with different label formatting.

## 3. Discussion

Here, we described an application of the StarSpace method to convert annotated genomic interval data into low-dimensional distributed vector representations. This builds on our previous work with embedding methods for genomic intervals [25], extending it to exploit accompanying metadata. We tested three different applications for joint embedding of data files and metadata. Each of these uses the joint representations of labels and genomic region sets in a shared space, making them comparable in low dimensions, which facilitates similarity search. This embedding space facilitates the calculation of vector distances for direct similarity comparisons and calculations between region sets, between labels, or even between a region set and a label. We also showed that the model can transfer annotations from a training set to new region sets without any labels. Not only does this make data search and retrieval more robust, but it can facilitate retroactive annotation of sparse or inaccurate metadata. To our knowledge, this is the first attempt to develop a system that leverages neural embedding methods to jointly embed genomic interval data with metadata for fast and effective dataset searching.

This approach provides a number of nice features for a search system. First, the region set search can query not only data used in training but also data in other region sets. Because we can use the trained model to convert new, unseen region sets to numerical vectors, a trained model can be used to search beyond the training data. This means no metadata information is required for the region sets to be included in the search, allowing us to search poorly annotated files. Second, this also means the database can be continuously updated to include new region sets to query against. Third, this approach is highly scalable as it is not dependent on region overlap analysis after the embeddings are computed. Fourth, it provides a natural similarity ranking system based on distances in the embedding space. Given these benefits, we propose this general approach could be used to build an impactful search engine that directly relates biological search terms to region sets regardless of the metadata annotating the original dataset. The user would communicate to the system by providing one or more labels within the metadata vocabulary, such as cell line, target antibody, or tissue type of interest. The output is one or more genomic interval sets that are biologically relevant to the provided query terms. Our proposed model has the potential to facilitate data annotation and metadata standardization as well as to assist biologists with analyzing similar experiments by identifying BED files that match a given query BED file.

Our models have several areas for future development. First, one limitation of our approach is that, like other NLP methods, it requires a shared vocabulary. New research in constructing such vocabularies for genomic interval datasets therefore promises to improve the accuracy of our method [33]. Second, we are also working on methods to evaluate the quality of the resulting embeddings, which will improve the quality of the models [34]. Finally, we will seek to expand the scope of natural language terms to enable a practical natural language search interface. In the future, we believe this approach could form the basis of a general-purpose genomic interval search framework capable of either natural language or interval-based queries. We also believe the annotation capability of this framework is a useful step forward for standardizing metadata annotations for diverse public datasets.

## 4. Methods

### 4.1. Training StarSpace Models

StarSpace is a general representation learning method that can embed different types of entities into an embedding space [30]. An entity could be anything. For example, in a text classification task, an entity could be a label or a sentence. This enables direct comparisons between different types of entities by learning embeddings of all the entities in the same space. In StarSpace, each entity is represented by a unique feature or a set of features: all from a fixed-length feature embedding dictionary. Specifically, we denote the dictionary as D, which is a D×d matrix. The *i*-th row $D_{i}$ of D represents a *d*-dimensional embedding of the *i*-th feature. Then, an entity a consisting of a set of features F is represented as $a=\sum_{i\in F}D_{i}$. In the context of joint learning of genomic interval sets and metadata, all the intervals and the labels of interval sets are the features that form the dictionary. Specifically, a label is an entity with a single feature, and an interval set is an entity with multiple features, where each interval in the set corresponds to a feature. The goal of StarSpace is to learn the feature embedding dictionary.

### 4.2. Tokenization Process

To map region set data to the format required for input into StarSpace, our initial step requires creating a consensus set of regions, which we define as the "universe of regions". This is achieved by tiling the reference human genome in intervals of 1000 base pairs, which collectively constitute this universe. To tokenize a given region set into the universe means we calculate the overlaps between each data point and the predefined universe. When an overlap is detected, the corresponding genomic tile is tagged as an existing token and is subsequently added to the document representing that specific data point. Each region set, or document, thus represents a collection of tiles for which the input data are mapped to the overlapping tiles in the universe. These regions are concatenated to form a document. The label of each document is appended at the end of this sequence, rendering the data prepared for subsequent processing using the StarSpace method to derive embeddings. These documents are formatted as required by StarSpace.

### 4.3. Models with Different and Combined Label Sets

A StarSpace model can be trained on one or several labels (Figure 1B–D). To test whether this could be used to accommodate different types of biological labels, we trained three StarSpace models: first, on the region sets labeled by only the antibody metadata; second, on region sets annotated by cell type metadata; and finally, on the antibody and cell type labels simultaneously. The model trained on both label types has the advantage of having a single embedding space for all the labels, which facilitates converting region sets to a numerical representation since the choice of the model is not dependent on the search term. However, the single-label models have a simpler learning task by having a reduced number of classes that the model is supposed to distinguish.

### 4.4. Training Procedure

Following the required format for the training phase of the StarSpace model, the labels (i.e., the metadata information for each region set) are appended to the end of each tokenized region set. For models with more than one label for each region set, we add all of them to the end of the document. We use the first training mode (trainMode = 0) of the StarSpace algorithm and cosine similarity as the loss function. This means StarSpace is jointly trained on both labels and documents such that label–document pairs are close to one another in the embedding space (Figure 1E).

Starspace learns the dictionary by comparing entities. Each entity is represented by its embedding computed from D. We sample pairs of entities containing both positive and negative pairs. A positive pair $\left(a,b^{+}\right)$ means entity a and positive sample $b^{+}$ are similar based on a similarity measure s(·,·). On the other hand, a negative pair $\left(a,b^{-}\right)$ means entity a and negative sample $b^{-}$ are dissimilar, and the value of $s(a,b^{-})$ is small. For each entity a, we sample a positive sample $b^{+}$ from the set of all entities to form a positive pair with a, and we sample *k* negative samples $b^{-}$ to form *k* negative pairs with a. Then, we minimize the following loss *L*:(1)$$\begin{array}{c} L=\sum_{a\in E}\sum_{b^{+}\in E_{a}^{+},b^{-}\in E_{a}^{-}}\ell \left(s\left(a,b^{+}\right),s\left(a,b_{1}^{-}\right),\ldots ,s\left(a,b_{k}^{-}\right)\right), \end{array}$$
where E is the set of all entities, and $E_{a}^{+}$ and $E_{a}^{-}$ are the positive and negative entity sets, respectively, for a. The loss function *ℓ* is a negative log loss of softmax, i.e., $\ell (s\left(a,b^{+}\right),s\left(a,b_{1}^{-}\right),\ldots ,$$s\left(a,b_{k}^{-}\right))=-\log\frac{e^{s(a,b^{+})}}{\sum_{i=1}^{k}e^{s(a,b_{i}^{-})}+e^{s(a,b^{+})}}$. For joint learning the embeddings of genomic interval sets and metadata, a is a genomic interval set, $b^{+}$ is a metadata label associated with a, and $b^{-}$ is some other label that is not associated with a.

### 4.5. GEO Projection

We downloaded 75,575 BED-like files from GEO using geofetch [32]. Of these, 32,174 annotate the *hg19* genome assembly according to the Genome_build column. Since our model was trained on hg19 data, we restricted the analysis to this subset.

### 4.6. Evaluation Metrics

#### 4.6.1. R-Precision

R-precision is used to evaluate the quality of *r* retrieved items for a given query. If we are given *r* items for a query, then the r-precision is defined as the proportion of the *r* items that are *relevant* [35]. To evaluate the overall quality of multiple queries, we average the individual r-precisions over multiple queries. If our method retrieves *r* region sets for the specific search term, then r-precision is the fraction of relevant region sets in the r-retrieved region sets.

#### 4.6.2. F1 Score

The F1 score [36] measures the performance of a classifier, with 1 being the best and 0 being the worst. It can be interpreted as a harmonic mean of the precision and recall, and thus, it can be called a balanced evaluation metric. The F1 score is defined as follows:(2)$$F1_{score}=\frac{2\times Precision\times Recall}{Precision+Recall}.$$ Precision is the fraction of correct predictions among all positive predictions and is defined as follows:(3)$$Precision=\frac{TP}{TP+FP}$$
where TP means true positive and FP means false positive predictions. Recall is the fraction of correct predictions among all positive samples and is defined as:(4)$$Recall=\frac{TP}{TP+FN}$$
where FN is the number of false negative predictions. In multi-class prediction, we adopt the micro-averaged F1 score, which is calculated by treating TP, FP, and FN as the sum of individual class TP, FP, and FN scores. For class *c*, its TP is interpreted as the number of samples that are predicted as *c* and are also from *c*; its FP is interpreted as the number of samples that are predicted as *c* but are from classes other than *c*; its FN is interpreted as the number of samples that are predicted as not *c* and are also from classes other than *c*.

#### 4.6.3. Mean Reciprocal Rank

Mean reciprocal rank (MRR) is an information retrieval measure that evaluates the performance of a retrieval system. MRR is defined as the average of the reciprocal ranks of all the queries. The reciprocal rank (RR) calculates the reciprocal of the rank at which the first relevant item was retrieved [35]. For example, in Scenario 1, given a query label, if the relevant region set is retrieved at rank 1, then the RR for this query is 1; if the set is retrieved at rank 2, then the RR for this query is 0.5. Formally, given a query item *q* from the set of queries Q, its RR is RR(q)=1/rank(q). The MRR for multiple queries is defined as follows:
(5)$$\begin{array}{c} MRR\left(Q\right)=\frac{1}{\left|Q\right|}\sum_{q\in Q}RR\left(q\right) \end{array}$$
where |Q| denotes size of Q.

## Figures and Tables

**Figure 1 bioengineering-11-00263-f001:**
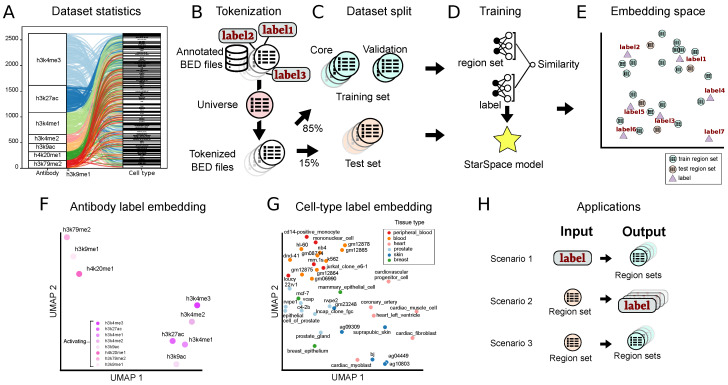
Overview of the approach and summary of results. (**A**) Alluvial plot showing the frequency of each class in the antibody and cell type data sets and their relationships to one another. (**B**) The first step is tokenization, in which annotated BED files are converted into a shared vocabulary using a region universe. (**C**) Tokenized files are divided into training core, validation, and test sets. (**D**) The Starspace model is trained using both the region set and label data. (**E**) The training procedure results in co-embedding space for both region sets and labels. (**F**) UMAP plot of the antibody label embeddings. (**G**) UMAP plot of the cell type label embeddings, colored by their tissue of origin. (**H**) We evaluated the model with three scenarios—1: retrieve region sets given a query label; 2: annotate a query region set with a label from the embedding space; and 3: retrieve similar region sets given a query region set.

**Figure 2 bioengineering-11-00263-f002:**
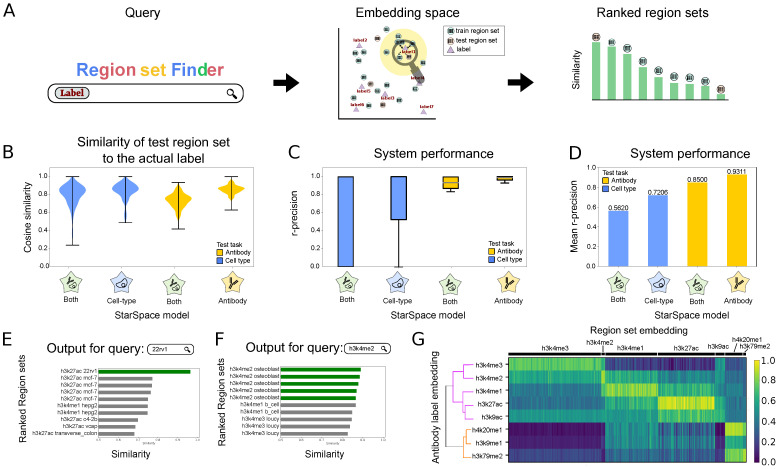
Description of and results for Scenario 1: retrieving relevant region sets given a query search label. (**A**) Overview of retrieval process. The search label is mapped to the embedding space, and nearby region sets are retrieved, returned, and ranked by distance to the query term embedding. (**B**) Distributions of cosine similarity of the test region sets to their true labels for 3 StarSpace models. (**C**) R-precision of the models trained on one or multiple labels for each search term in the antibody or cell type labels. (**D**) Average r-precision of each model. (**E**) Example of 10 most similar region sets for the query term 22rv1 from cell type label group. Green bars indicate a perfect match. (**F**) Example of 10 most similar region sets for the query term H3K4me2 from antibody label group. Green bars indicate a perfect match. (**G**) Heatmap showing the similarity between the antibody label embeddings and the embeddings of the region sets in the test dataset. The dendrogram depicts relationships among the label embeddings.

**Figure 3 bioengineering-11-00263-f003:**
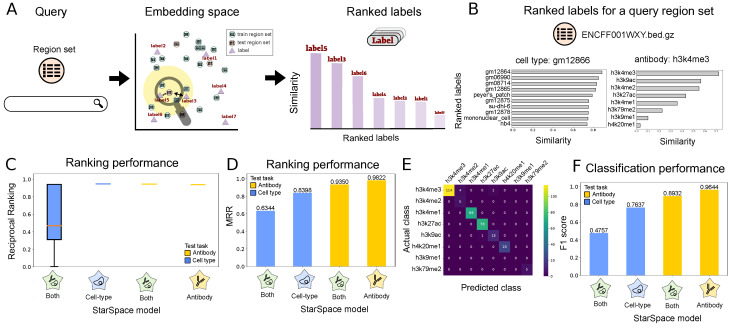
Description of and results for Scenario 2: annotating query region sets. (**A**) Overview of Scenario 2: first, the query region set is mapped to the embedding space; then, the labels are ranked based on their similarity to the region set in the embedding space. (**B**) The ranked antibody (left) and cell type (right) labels based on the similarity metric for the *ENCFF001WXY.bed* query region set. (**C**) Reciprocal ranking for the models on one and multiple labels. (**D**) Mean reciprocal ranking for the models on one and multiple labels. (**E**) Confusion matrix of the true class of antibody labels and predicted antibody by our method. (**F**) The classification performance (F1 score) of models trained on one and multiple labels.

**Figure 4 bioengineering-11-00263-f004:**
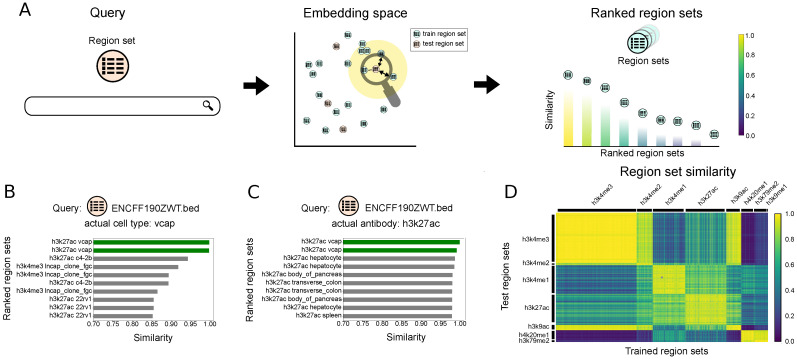
Description of and results for Scenario 3: Retrieving region sets similar to a query region set. (**A**) Overview of Scenario 3: the query region set is mapped to the embedding space, and the retrieved region sets from the database are ranked based on the their similarities to the query region set embedding. (**B**) Ten most similar region sets to the query region set, ENCFF190ZWT.bed, for model trained on antibody labels. (**C**) Ten most similar region sets to the query region set, ENCFF190ZWT.bed, for model trained on cell type labels. (**D**) The heatmap plot of the similarity between the test region set embeddings and the embeddings of the region sets in the training dataset. The model is trained on the antibody labels.

**Figure 5 bioengineering-11-00263-f005:**
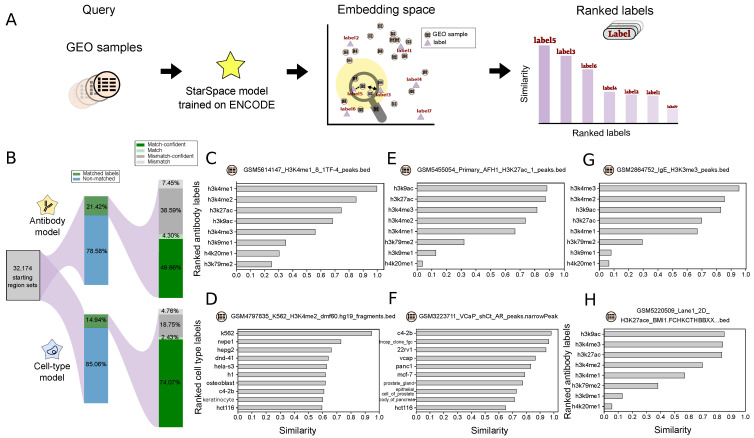
Application of our model for annotating a new dataset: (**A**) Overview of the approach: query region set from GEO database is mapped to the embedding space, then labels are ranked based on their similarity. (**B**) Statistics of the collected data and the percentage of the match/mismatch prediction. (**C**–**H**) Example search results for the given query file; panels (**C**,**D**) show examples of correctly predicted labels; panels (**E**,**F**) show examples of near misses; panels (**G**,**H**) show examples of results for non-standard labels.

## Data Availability

The training code and models used can be found at: https://github.com/databio/geniml (accessed on 27 February 2024).

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
