# Peer review of "Joint Representation Learning for Retrieval and Annotation of Genomic Interval Sets"

_bioengineering, 2024, doi:10.3390/bioengineering11030263_

Round 1

Reviewer 1 Report

Comments and Suggestions for Authors

Summary and Strengths:  

The authors propose a  fast search method for large-scale genomic interval data. 

This method trains numerical embeddings for region sets and their metadata labels,  and captures similarity between region sets and their metadata in a low-dimensional space.  

This method successfully solves 3 information retrieval tasks using embedding distance computations: retrieving region sets related to a user query string, suggesting new labels for database region sets, and retrieving database region sets similar to a query region set. 

The experimental results demonstrate the effectiveness of the proposed method. 

Weakness:

1. The StarSpace method should be further clearly formulated, e.g., how the original genomic interval data is embedded into low-dimensional formation, how the similarity is calculated. 

2. StarSpace is an old method presented in 2018. So, why such method is taken to handle the genomic search task? The suggestion is to theoretically explain why it is specific to this task, or compare it with other new methods in practice. 

Author Response

Review response uploaded as PDF.

Reviewer 2 Report

Comments and Suggestions for Authors

Retrieval and annotation of genomic interval sets is a challenging task but has significant value for researchers. In this manuscript, the authors leveraged deep learning approaches to learn the representation of genomic intervals and their  annotations. They reported the detailed implication and used three cases to demonstrate the utility of their search system. This method overcomes the limitations of traditional methods by the combination of sequence information and metadata labels thus have the potential for inferring biological functions of those genomic intervals.  The authors also point out the future directions for further improvement which I am very interested. Overall this is a very good paper and though the performance may be limited by current available datasets for training.

Author Response

Review response uploaded as PDF.

Reviewer 3 Report

Comments and Suggestions for Authors

The biggest issue with this article is the lack of availability of any code. Where is the code? How can researchers reproduce your work? Please put your code on Github with a README file immediately.

Author Response

Review response uploaded as PDF.

Round 2

Reviewer 3 Report

Comments and Suggestions for Authors

The code has now been included in the Github page. Thank you for making this revision.